# Increasing the Beam Width and Intensity with Refraction Power Effect Using a Combination of Beam Mirrors and Concave Mirrors for Surgical-Fluorescence-Emission-Guided Cancer Monitoring Method

**DOI:** 10.3390/s24175503

**Published:** 2024-08-25

**Authors:** Jina Park, Jeongmin Seo, Kicheol Yoon, Sangyun Lee, Minchan Kim, Seung Yeob Ryu, Kwang Gi Kim

**Affiliations:** 1Premedicine Course & Department of Medicine, College of Medicine, Gachon University, Incheon 21565, Republic of Korea; jnpr444@gmail.com (J.P.); jungmin00010@naver.com (J.S.); kcyoon98@gachon.ac.kr (K.Y.); 2Medical Devices R&D Center, Gachon University Gil Medical Center, Incheon 21565, Republic of Korea; kormd98@naver.com (M.K.); wyverns0723@gmail.com (S.Y.R.); 3Department of Radiological Science, Dongnam Health University, Suwon 16328, Republic of Korea; leesy2024@dongnam.ac.kr; 4Department of Biohealth & Medical Engineering Major and Biomedical Engineering, Gachon University, Seongnam-si 13120, Republic of Korea; 5Department of Health Sciences and Technology, Gachon Advanced Institute for Health Sciences and Technology (GAIHST), Gachon University, Incheon 21565, Republic of Korea

**Keywords:** LED, refraction power, mirror, concave lens, fluorescence emission

## Abstract

The primary goal during cancer removal surgery is to completely excise the malignant tumor. Because the color of the tumor and surrounding tissues is very similar, it is difficult to observe with the naked eye, posing a risk of damaging surrounding blood vessels during the tumor removal process. Therefore, fluorescence emission is induced using a fluorescent contrast agent, and color classification is monitored through camera imaging. LEDs must be irradiated to generate the fluorescent emission electromotive force. However, the power and beam width of the LED are insufficient to generate this force effectively, so the beam width and intensity must be increased to irradiate the entire lesion. Additionally, there should be no shaded areas in the beam irradiation range. This paper proposes a method to enhance the beam width and intensity while eliminating shadow areas. A total reflection beam mirror was used to increase beam width and intensity. However, when the beam width increased, a shadow area appeared at the edge, limiting irradiation of the entire lesion. To compensate for this shadow area, a concave lens was combined with the beam mirror, resulting in an increase in beam width and intensity by more than 1.42 times and 18.6 times, respectively. Consequently, the beam width reached 111.8°, and the beam power was 13.6 mW. The proposed method is expected to be useful for observing tumors through the induction of fluorescence emission during cancer removal surgery or for pathological examination in the pathology department.

## 1. Introduction

Medical science technology and diagnostic medicine are advancing rapidly. Due to these developments, the 5-year survival rate (>91%) is increasing due to early cancer detection and surgical treatment [1]. Highly invasive malignant tumors metastasize due to cancer fragments remaining after tumor resection and have a high cancer recurrence rate, so the maximum goal of surgery is complete tumor resection. However, numerous blood vessels exist inside the tumor, and the colors of the tumor and blood vessels are similar to each other, making it difficult to observe with the naked eye [2]. Therefore, it is difficult to distinguish the boundary between the tumor and blood vessels, and blood vessels may be damaged during the tumor removal process, or tumor resection may be inadequate. To determine whether the tumor has been resected, a fluorescent contrast agent is injected into the lesion, and when an LED with an excitation wavelength of 405 nm is irradiated to the lesion, the tissue stained with the fluorescent contrast agent emits 530–680 nm due to a chemical reaction. A fluorescent emission with a wavelength occurs [3]. Fluorescent emission wavelengths have colors for each spectrum. Therefore, when monitored through camera imaging, lesions are expressed in color [4]. This observation can be made using an operating microscope. The surgical microscope is equipped with LEDs. The lesion area is >5 cm [4]. The beam width of the LED is around 20°, making it difficult to irradiate the entire lesion. In addition, due to the working distance (WD: 15–30 cm) between the LED and the lesion, the beam power of the LED is consumed by the distance, which weakens the electromotive force of fluorescence emission from the lesion, making it difficult to distinguish and observe the color of the lesion [3]. In addition, when the beam width is increased, the beam power of the LED decreases by the amount of the beam width increase, so there have been cases where methods of increasing the beam width and beam intensity using multiple LEDs have been studied [5,6,7,8,9,10,11,12,13]. However, as the number of LEDs increases, power consumption and unit cost increase, and breakdowns increase due to vulnerability to heat [8,14]. Due to this uneconomical nature, there has been research on how to increase the beam width and beam power with one LED [4]. The results of [4] are very groundbreaking and useful in clinical settings. However, the beam distribution shape is analyzed to be diamond-shaped. It is clear that the beam width is wider, and the intensity has increased compared to the existing LEDs [4]. However, because the beam intensity disappears at the edge of the beam distribution due to the effect of refraction, the expected result of the beam being distributed all the way to the end leaves something to be desired. Therefore, in order to complement these results, there is a need to overcome this unfortunate phenomenon by proposing a refractive power effect method.

This paper proposes a method to further widen the beam width while increasing the beam intensity using beam refraction power. The proposed method uses a concave lens to maximize the refractive power effect and obtains optimal results by complementing the beam width reduction area and the beam intensity reduction area (edge).

## 2. Analysis of Refraction Power Effect to Increase Beam Intensity and Beam Width

### 2.1. Improvement of Beam Width and Intensity Increase to Induce Fluorescence Emission Throughout the Lesion

A method to observe the lesion state in color through the induction of fluorescence emissions is to inject a fluorescent contrast agent into the lesion and stain the lesion with fluorescence, as shown in Figure 1 [9,10]. LED is irradiated to the stained lesion to induce fluorescence emissions through a chemical reaction. Since the fluorescence emission wavelength contains the unique color of the spectrum, when photographed with a camera, the lesion is displayed in color on the monitor [11,12,13]. At this time, the LED that irradiates the beam to the lesion has a beam width of 10° to 20°, as shown in Figure 2 [4,15,16,17,18]. This beam width has limitations in irradiating the entire lesion. The beam width of the LED must be increased to irradiate the entire lesion. In addition, the beam power of the LED is lost due to the constant WD between the LED and the lesion, and because the beam power has the characteristic of becoming weaker as the beam width becomes wider, the electromotive force that induces fluorescence emission may be weak. Therefore, after obtaining refraction and reflection effects through a beam mirror, a quasi-symmetrical irradiation method was used to obtain a result in the form of a diamond-shaped beam width as shown in Figure 2, providing an excellent research example that revealed the effect of increasing the beam width and beam intensity.

According to the research results, the beam width (43°) and beam intensity were effectively increased. In addition, the design method is very simple without a separate semiconductor process, so it is expected to be applicable in clinical settings. However, through the reflection and refraction effect of the beam using a beam mirror, shaded areas are created in the -i_n_ and i_n_ areas due to the diamond shape. These shaded areas make fluorescence emission difficult. Therefore, we propose a way to overcome this problem by supplementing the shaded area to enable sufficient fluorescence emission from lesions with a large area.

### 2.2. Method for Deriving Refractive Power Phenomenon

A concave lens is used to widen the beam width of the LED. When using an LED and a concave lens, the beam increases from 78.8° to 109°, and the power increases from 0.73 mW to 10.9 mW, as shown in Figure 3a. If the beam mirror and concave lens are connected together, as shown in Figure 3b, the beam width and beam intensity increase significantly. Therefore, it is necessary to analyze the refractive power effect to further increase the beam width and beam intensity by combining a beam mirror and a concave lens [19].

The concave lens consists of a projection surface (R_2_) and a transmission surface (R_1_), as shown in Figure 3c. The LED beam that is refracted and collected through the beam mirror passes through R_2_ at a constant distance (d_1_) and width (f_1_) to the concave lens. When the beam (f_c_) passes through the lens, the beam spreads by d_2_ through the beam’s refraction (*n*) according to the difference in curvature (θ) of the lens, as shown in Equation (1).
(1)$$\theta =\eta -\left(f_{1}\frac{R}{h}\right)$$

In order to spread the LED beam (f_c_) in the concave lens, the distance between R_1_ and R_2_ must be thick, as shown in Figure 4 and Equation (2), and R_1_ must be wider than R_2_ by R_1_ > R_2_ [19]. Therefore, the magnification (R) when the beam is transmitted must be at least 1.2 (>1.2). Since θ is the angle at which the beam obtained through the beam mirror spreads to the concave lens is 0°, assuming that tan θ is ignored, the magnification (R) can be obtained through $\left(\frac{1}{R_{1}}+\frac{1}{R_{2}}\right)$. The actual beam width is interpreted as in Equation (3).
(2)$$\frac{1}{m}=tan\theta \left(\frac{1}{R_{1}}+\frac{1}{R_{2}}\right)=\left(\frac{1}{R_{1}}+\frac{1}{R_{2}}\right)$$
(3)$$R={tan}^{-1}\left(\frac{\frac{f_{1}}{1000}}{17.5}\right)\;\;\;\;\;@\;R_{2}=R_{1}\times R$$

Therefore, when the beam is gathered in a narrow R_2_, the energy of the beam is concentrated (*x*) as shown in Equation (4), increasing the intensity of the beam, and the increased intensity of the beam spreads a certain area in R_1_ as shown in Figure 5a [19]. Therefore, as shown in Equation (5), the power of the beam increases as its width increases, which is defined as the refraction power of the beam as shown in Equation (5) [19].
(4)$$R=arctan\left(\frac{R}{d_{1}}\right)$$
(5)$$Beam\;angle({}^{\circ})\approx 2\times \tan^{-1}\left(\frac{R}{d_{1}}\right)$$

Refraction power means that the beam is refracted according to the size of the lens, increasing the beam width and intensity. In other words, the effects of absolute power increase (R_1_ > R_2_) and beam width expansion (R) are used. Therefore, the intensity of the power reaching directly from the LED to the concave lens (R_1_) and the power generated at the location where refraction begins through the concave lens overlap each other, and the beam width is expanded.

Therefore, the refraction power has the characteristic that the beam width widens as the power of the LED increases uniformly, as shown in Figure 5. General LEDs have a relatively narrow beam width, and the beam intensity becomes weaker from the center. However, if the experiment with the spread of the beam through monitoring by configuring an LED, a concave lens, and a video camera, as shown in Figure 5a,b, we can obtain results on the change in the refraction power of the beam.

Using refractive power has the advantage that the intensity of the beam does not change but rather increases uniformly, and the beam width widens. However, for LEDs using beam mirrors, the beam width and beam intensity increase uniformly. The reason is that if a concave lens is used, the effect can be increased by more than *N* times, and if the refractive power effect is used, the beam width and intensity can be increased.

## 3. Experimental Methods and Results

### 3.1. Experimental Environment Configuration

In order to obtain the results of the refractive power effect by connecting the beam mirror and the concave lens, a square-shaped box, as shown in Figure 6, is produced directly using 3D printing to create optimal conditions for the experiment. From the figure, the structure is classified into layers A, B, C, and D, and in D, j has a diameter of 5 cm. h and y are each 5 cm, and k and m are each 16.5 cm.

Efforts are made to obtain experimental results by reconfiguring the rectangular box similar to [4] by inserting one LED and four beam mirrors. Additionally, a near-infrared (NIR) camera is installed inside the square-shaped box. Since a concave lens is inserted horizontally at the bottom of the beam mirror to obtain a refractive power effect, the shape of the structure and WD may be expressed slightly differently from [4]. Additionally, when a beam mirror and a concave lens are inserted together, the beam width shape changes, so the WD is bound to change.

The distance of the mirror position away from the LED is 13 cm, a concave lens is installed horizontally on the lower part of the mirror, and the bottom surface 5 cm away is installed separately to measure power. A phantom or lesion capable of fluorescence emission can be placed on the bottom surface. When a beam is emitted from an LED, the beam is divided into three parts and distributed toward both mirrors and the concave lens, as shown in Figure 7. Therefore, as the beam width of the beams that are reflected and refracted from the mirror increases, the beam width power will also increase. At this time, the beam that directly reaches the concave lens and the power that is reflected and refracted through the mirror and reaches the concave lens are combined, and then all of the beams will pass through the concave lens. For beams that pass through a concave lens, the intensity of the beam increases, and the beam is evenly distributed to the shaded area, resulting in a wide increase in beam width.

Figure 8a shows a square box in which a beam mirror, concave lens, and LED are installed to obtain the experimental results. A power sensor is applied to test the beam power. The mirror used for the experiment is a total reflection mirror, and the concave lens and mirror are directly connected in the vertical and horizontal directions. Figure 8b is a photograph of a power sensor with an optical filter attached. The power sensor has a built-in mode that can select a wavelength band of 405 nm. Therefore, the sensor has the performance to detect power corresponding to 405 nm. However, if the surrounding fluorescent light or another light source wavelength is detected by the power sensor along with the power corresponding to 405 nm, accurate power may not be generated. Therefore, a bandpass filter is installed on the power sensor. The bandpass filter is self-made and has a transmittance and reflectance of approximately 98.6% and 0.12% at 405 nm (±10 nm). The power loss reaching the power sensor due to this transmittance and reflectance is negligible. When irradiating a 200 mW LED (@ working distance of 18 cm), the power reaching the power sensor is classified into two categories. The power before applying the optical filter to the sensor was measured to be 13.72 mW, and the power when the optical filter was applied was measured to be 13.6 mW. That is, the difference in power loss before and after installing the filter is 0.12 mW.

To obtain experimental results through fluorescence emission, a phantom was manufactured as shown in Figure 7. The phantom model has various curves, providing optimal conditions for inducing light reflection.

Phantom fabrication is shown in Figure 9. Mix 3 cc of Fluorescein sodium 10% for injection with 97 cc of sterile distilled water. The mixed liquid is mixed with 20 mL of silicone rubber liquid to form a liquid. The mixed liquid is injected into a silver cell dish, filled, and then heated to a temperature of 60 degrees for about 6 h. The heated liquid turns into a solid, completing the production of a solid phantom. The size of the phantom is similar to the size of the specimen extracted from the body.

### 3.2. Experimental Method and Result Analysis

The LED radiates a beam with an excitation wavelength of 405 nm and a power of 200 mW to the target. As shown in Figure 10, the beam width and beam power when only the LED is irradiated without a mirror and a concave lens, the beam width and beam power of the LED when only the beam mirror is present without a concave lens, and when both the beam mirror and the concave lens are installed. Differences in changes in beam width and beam power can be observed. These results are presented in Table 1.

A comprehensive analysis is performed on the results, as shown in Figure 11. The beam width is compared, and the difference is analyzed. When only the general LED (M (−)//L (−)) without a beam mirror and concave lens was irradiated and the LED (M (+)//L (+)) equipped with a beam mirror and concave lens was compared, the beam width differed by a factor of 1.42. In other words, the LED (M (+)//L (+)) equipped with both a beam mirror and a concave lens had a wider beam width than the LED (M (−)//L (−)) irradiated without a beam mirror and concave lens. At this time, when the beam width of the general LED (M (−)//L (−)) without a beam mirror and concave lens and the LED-equipped only with a beam mirror (M (+)//L (−)) were analyzed, a difference of 1.13 times occurred. Finally, the LED (M (+)//L (+)) equipped with both a beam mirror and a concave lens showed a beam width expansion of more than 1.42 times. A comparative analysis of the beam power is made. When the beam power of a general LED (M (−)//L (−)) without a beam mirror and a concave lens and an LED (M (+)//L (−)) equipped only with a beam mirror was analyzed, a difference of 1.49 times occurred. When only a general LED (M (−)//L (−)) was irradiated, and the LED (M (+)//L (+)) equipped with a beam mirror and a concave lens were compared, the beam power differed by 1.86 times. Finally, the LED (M (+)//L (+)) equipped with both a beam mirror and a concave lens showed an increase in beam power of more than 1.86 times (PW4/PW1 = 1.86 times).

As shown in Figure 12, the ability of the beam width to irradiate the entire phantom is tested through LED irradiation. The measure of whether fluorescence emission can supply sufficient power is to review the satisfaction of the results by measuring the LED beam width and emission power in a state where light reflection is possible. Therefore, we analyze the state in which the LED beam width can sufficiently irradiate the entire phantom and determine the performance of whether the beam width has been increased and the possibility of sufficient fluorescence emission by increasing the beam intensity.

In the absence of a mirror, fluorescence emission is impossible, but PW2 in Table 2 is capable of fluorescence emission. At this time, the fluorescence emission range was observed to be 28.4 cm in diameter at the brightest state. However, under the irradiation conditions of PW4, the fluorescence emission range was observed to be expanded to about 35.1 cm. This is because of the difference in beam irradiation range between the LED without a mirror or concave lens and the LED with a mirror and concave lens.

At this time, there is a way to obtain results through video recording for the fluorescence emission possibility test, and another way is to measure the fluorescence emission power to obtain the fluorescence emission possibility numerically. Therefore, we conducted an experiment on the fluorescence emission power that occurs when the power corresponding to PW1 to PW4 reaches the fluorescently dyed material, as shown in Table 1. For the fluorescence emission measurement, we attempted to measure it by installing a fluorescence emission phantom on the power sensor.

PW1 (without mirror and concave lens) is not capable of fluorescence emission. However, since fluorescence emission is possible from Pw2 (with a mirror and without a concave lens), we attempted to compare PW2 to PW4 (with a mirror and concave lens).

Regardless of the working distance value, when the irradiation power is 200 mW at 405 nm, the power reaching the fluorescent dyed material must be at least 0.5 mW or more so that the fluorescence wavelength of 530–560 nm can be emitted from the fluorescent material [4,20]. Therefore, we measured the fluorescence emission power according to the power (PW1 to PW4) reaching the fluorescently dyed material when the investigation wavelength, working distance, and investigation power were 405 nm, 18 cm, and 200 mW. For the fluorescence emission measurement, we used a phantom dedicated to fluorescence emission. This fluorescence emission power was analyzed, as shown in Table 1, and it was observed that fluorescence was emitted from both PW2 and PW4. Figure 13 is a figure related to the experimental environment configuration for measuring fluorescence emission power. PW2 is the start of fluorescence emission, and although fluorescence emission has started, the fluorescence concentration is not bright, which may make the lesion observation field dark or difficult to see in clinical experience. However, in the case of PW4, since there is a difference of 1.31 times compared to the fluorescence start power (0.5 mW), a considerably bright fluorescence can be emitted, so the lesion observation field can be sufficiently secured. The reason is that when the fluorescence emission power is 4 mW, the fluorescence emission shape is clearly visible, and when the irradiation power (PW4) is 13.6 mW, the fluorescence emission power was measured to be 6.2 mW, so it was possible to record a value of at least 1.5 times [21].

## 4. Discussion

The method for increasing beam intensity and beam width did not require separate circuit configurations or new semiconductor processes. Methods to increase beam width and beam intensity in commercially available LEDs will require new semiconductor processes. Therefore, it is believed that a new semiconductor process will be needed every time the LED shape and wavelength changes. If the proposed method is applied every time the LED is changed, both the beam width and intensity can be increased, making it possible to sufficiently apply it to various types of LEDs.

Based on these analyses, we conducted experiments on the uniform increase in beam width intensity and beam intensity splitting in terms of various wavelength bands of LEDs applied in the medical fluorescence emission-induced lesion observation method, as shown in Figure 14 and Table 2. The wavelength bands are 405 nm, 505 nm, and 780 nm, and the irradiation power was equally set to 200 mW. In addition, the working distance was equally determined to be 18 cm. The experiments were classified into states with and without inserting a beam mirror and a concave lens, and the beam width, beam irradiation range, fluorescent dye reaching power, and fluorescence emission power were measured.

In general, mirrors include regular mirrors and total reflection mirrors [4]. We use a total reflection mirror. Since regular mirrors have a refractive loss (>0.35%), they are affected by LED beam power loss and beam width loss. However, a total reflection mirror has minimal refraction loss. Since the proposed method was tested using a phantom, it is necessary to obtain verification results of clinical suitability through future animal experiments. Therefore, animal experiments to evaluate performance reliability are planned as future research projects.

If we discuss the results of Figure 10, the edges of the beam irradiation shape are spread out, or a shadow is generated. This spreading phenomenon or shadow occurs when there is neither a mirror nor a concave lens (PW1) when the LED is spreading. The characteristic of the LED is that the beam intensity is weakened, and the beam is spread out at the same time. Also, PW2 is a state where only a mirror is installed without a concave lens. At this time, the beam is refracted and reflected in the area where the mirror exists, and the beam is focused. However, since there is no concave lens, the LED beam is strongly irradiated due to refraction, and a phenomenon occurs where it spreads like a shadow at the edge of the irradiation area. PW3 increases the beam width because there is only a concave lens without a mirror. Therefore, the beam spreads out as much as the beam width increases, so the occurrence of a shadow shape is minimized. However, PW4 can minimize the shadow or spreading phenomenon because it gathers the beam while increasing the beam irradiation intensity and beam width with both a mirror and a concave lens present. The focus of this study is the result of PW4. It is necessary to collect results on the phenomenon that occurs when both a beam mirror and a concave lens are present. The main purpose of this study is to increase the beam width and intensity through a single LED.

Of course, we can solve the brightness difference, shadow, or spreading phenomenon at the edge by using four LEDs [20,22]. In particular, there is a method to increase the beam intensity and beam width by increasing the number of LEDs, such as FLUOBEAM^®^ LX (GETIGE) manufactured by Fluoptics, Grenoble, France. The advantage is that we can observe the lesion widely and sharply. It is also a very innovative device that observes the blood circulation status of blood vessels and lymph nodes using an indocyanine green fluorescent contrast agent. However, as the unit price increases, the LED circuit configuration becomes complicated, and the power consumption of the LED increases. However, if we connect the LEDs in parallel and use the negative feedback gain increase method in the drive circuit, the circuit configuration can be simplified, and the power consumption can be minimized. If we connect the LEDs in parallel and use the negative feedback gain in the drive circuit, the power consumption can be minimized. If these rotate it using four LEDs, it seems that we can solve the problem as we suggested. It is judged that these methods can obtain positive results through future research.

Table 3 shows the differences and superiority of the results when compared with existing research results. When the WD is close, the beam power of the LED will be relatively low. However, the power loss to reach the target will not be significant. The power that a laser reaches the target is relatively higher than that of an LED, so even if only one laser is used, the power is bound to be high [23]. However, because lasers generate high heat, power consumption and heat generation increase. Lasers have a relatively narrow beam width and strong straightness compared to LEDs, so they have excellent fluorescence emission [21]. However, there will be the inconvenience of having to install a separate expander to irradiate the beam to the entire lesion [21,24,25]. In addition, if an expander is installed, there may be a loss of the beam (1.0–4.0%) due to the internal resistance of the lens [26].

However, it should be noted here that the judgment that the higher the power of the LED, the better the performance is a problem that needs to be analyzed again [27,28,29]. Since it is important to have a sufficient power level to enable fluorescence emission, it must be carefully determined that the wider the beam width, the better the performance. The reason is that when the beam is irradiated to the entire lesion, it must have the ability to generate sufficient fluorescence emission throughout the lesion, so the wider the beam width, the better the evaluation. In particular, we present analysis results showing that an LED with a wide beam width from one LED may have better performance than other methods.

This study expanded the beam width and intensity by increasing the refractive intensity using a beam mirror and a concave lens. This result enables beam irradiation to the entire lesion and induces fluorescence emission without adjusting the beam direction angle.

If an LED module with an integrated beam mirror and concave lens is connected to the end-effector using a robot manipulator to adjust the beam irradiation angle, the beam irradiation angle and working distance can be sufficiently adjusted [10].

Therefore, there is a need for continuous research and development in the future, and it is expected that these results can contribute to medical fluorescence observation and the development of medical robots. These results are indicated in Figure 15 and pink in the discussion. 

Finally, the wavelength band of 405 nm is used in the medical field for medical diagnosis sensors, biometric recognition sensors, diagnostic environment management, observation of lesions induced by fluorescence emission, and sterilization [30,31]. The reason is that if it is directly irradiated to the skin or eyes for a long time, it can cause damage, so it is handled quite meticulously to kill most bacteria or viruses [30]. In particular, the 405 nm wavelength band used in high-power LEDs requires heat management due to high power consumption [32,33,34]. LEDs with high emission power consume a lot of power, so the heat increases proportionally [32]. This heat reduces the performance of the LED and also shortens the lifespan of the LED [33]. In addition, the LED module may fail due to heat damage [34]. At this time, as shown in Figure 16, a temperature controller module is inserted to control the load current whenever the heat increases and the LED module is designed to transfer heat to the metal by installing a heat sink. Therefore, the LED module starts to generate heat and increases up to a maximum of 55.767 °C, but since a heat sink is installed, the heat is reduced to 53 °C. At this time, since some of the heat is cooled through the heat sink, the boundary between the heat sink and the module is observed to be approximately 49 °C.

## 5. Conclusions

In the process of increasing the beam width and beam irradiation power of the LED used to observe lesion color distinction through fluorescence emission, a diamond-shaped beam was generated due to the refraction and reflection characteristics of the beam mirror, and a shadow area was analyzed. Therefore, the shading area is supplemented to maximize beam irradiation, and the beam width and beam irradiation are further increased. To obtain these results, a concave lens was used, and the refractive power effect was discovered. This refractive power effect fills the shaded area with light to maximize beam width and beam intensity. Therefore, sufficient LED beam irradiation is possible for lesions with a large area, making it easy to observe the fluorescence emission of the entire lesion.

It does not require changing circuit characteristics or semiconductor processing and can be additionally installed on existing LEDs, making it quite easy to design or apply. The fact that beam width and beam intensity can be increased with one LED is seen as a significant advantage in itself. It is expected that it will be useful in departments specializing in surgery and departments specializing in procedures and diagnosis because it can be evaluated for clinical value by making it smaller and approaching animal testing. Animal experiments have plans for future research projects.

## Figures and Tables

**Figure 1 sensors-24-05503-f001:**
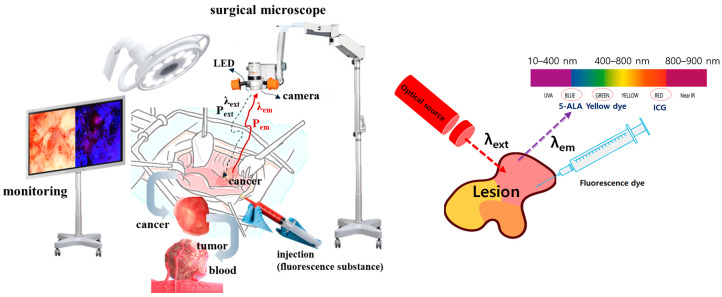
Fluorescence emission guided cancer removal observation during the cancer surgery (Monitoring of lesions by inducing fluorescence emission after injection of fluorescent contrast agent and irradiation with light source at the surgical site).

**Figure 2 sensors-24-05503-f002:**
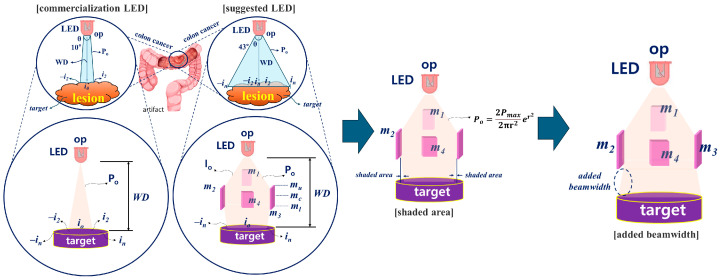
Conceptual diagram for increasing beam width and overcoming the occurrence of shadow areas using beam mirrors (Analysis of LED light source intensity difference before and after beam mirror installation).

**Figure 3 sensors-24-05503-f003:**
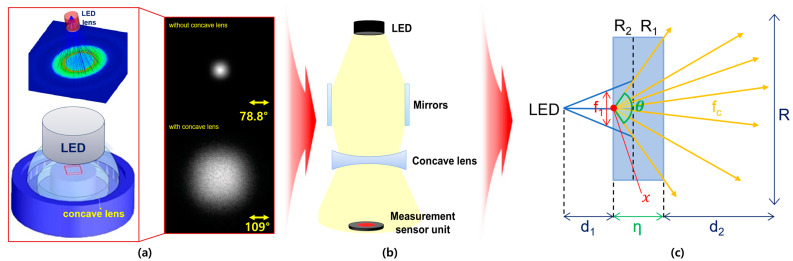
Analysis of refractive power effect with concave lens (**a**) simulation results (**b**) structure (**c**) beam diffusion (When a concave lens is mounted, refractive power effect occurs and beam intensity increases).

**Figure 4 sensors-24-05503-f004:**
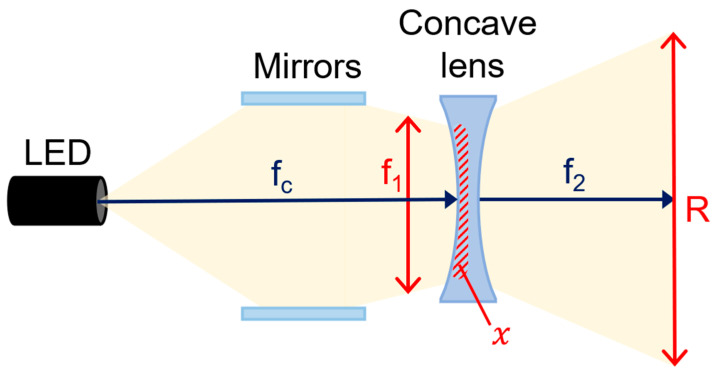
Structure of the refractive power effect using combination of beam mirror and concave lens (When a beam mirror and concave lens are mounted on an LED, refractive power effect occurs—beam intensity and beam width uniformity increase).

**Figure 5 sensors-24-05503-f005:**
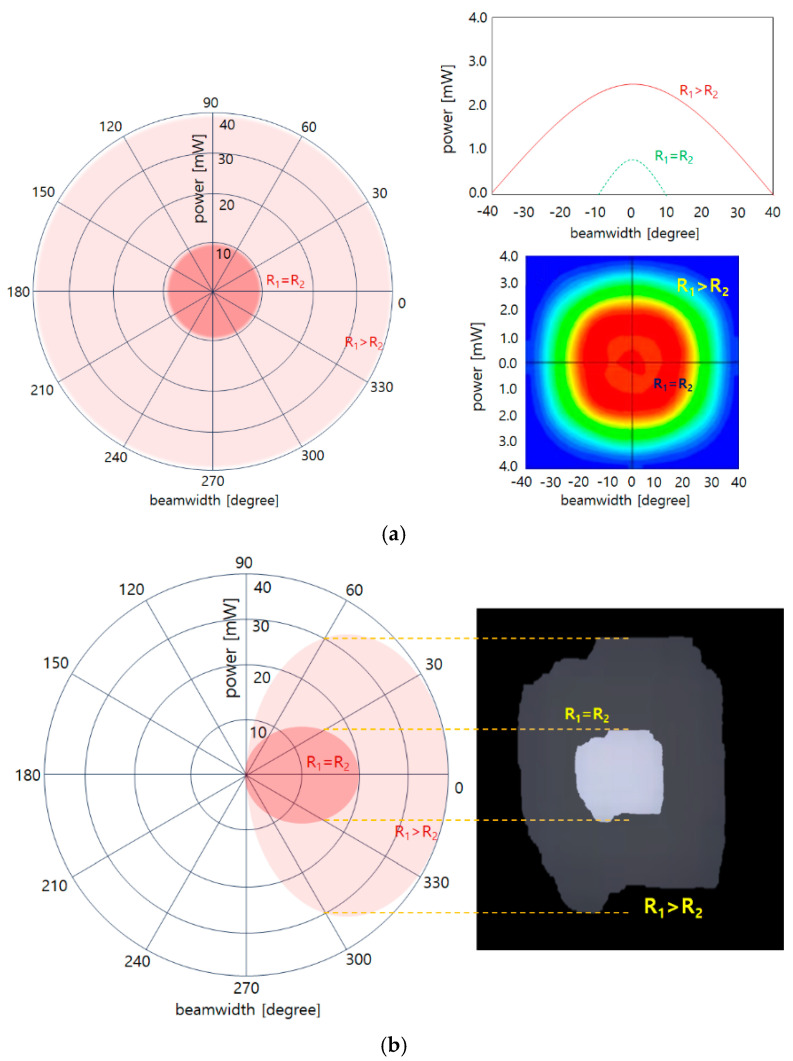
Simulation results for comparison of beam power and beam width with general irradiation and reflective power (**a**) front view and polar chart simulation (**b**) side view and beam diffusion effect simulation (beamwidth increase analysis is possible through polar charts).

**Figure 6 sensors-24-05503-f006:**
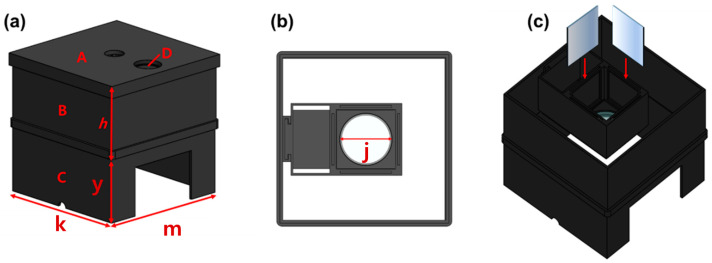
Experimental environment configuration using 3D printing technology (production) (**a**) entire structure (**b**) top view (**c**) mirror insertion and side view (When external light penetrates, the black color absorbs it and blocks the LED light from leaking out to the outside, so the material was decided to be black).

**Figure 7 sensors-24-05503-f007:**
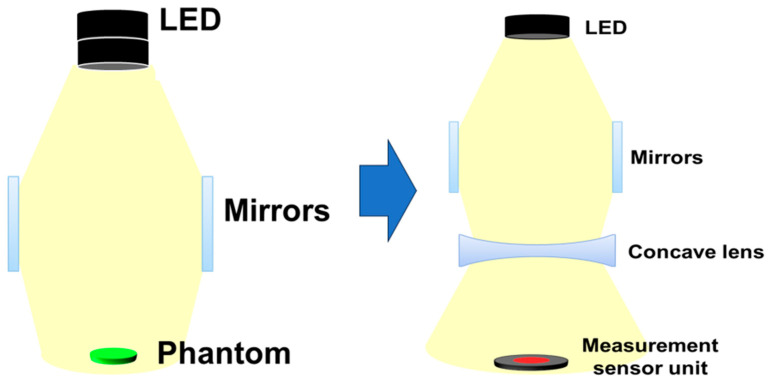
Configuration diagram for experiment (structural diagram to analyze the difference in beam irradiance when the beam mirror is mounted and when both the beam mirror and concave lens are mounted).

**Figure 8 sensors-24-05503-f008:**
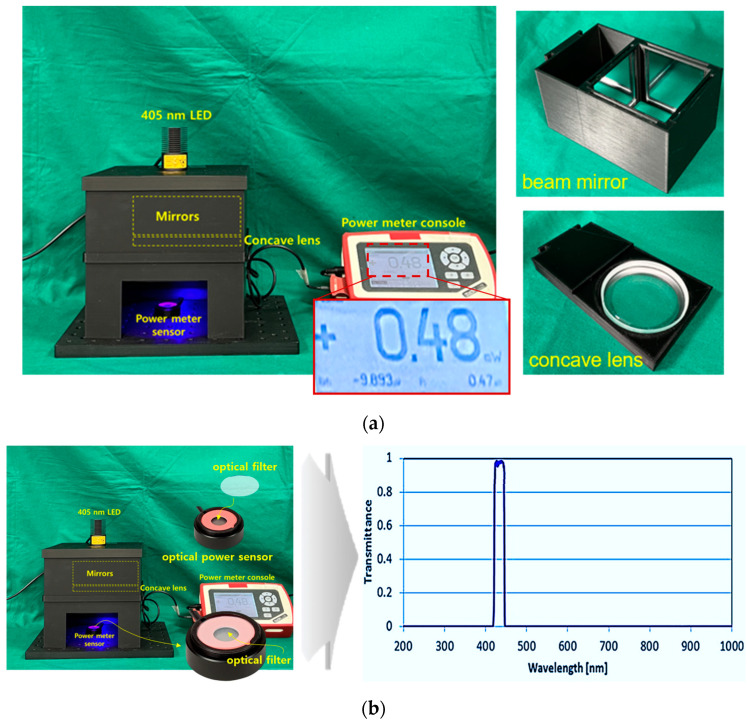
Results of experimental environment production (**a**) fabrication of experimental apparatus using 3D printing (**b**) optical filter mounted on power sensor and wavelength measurement results (LED, beam mirror, concave lens all mounted and fixed frame manufactured using 3D printer—power sensor mounted to measure LED beam intensity).

**Figure 9 sensors-24-05503-f009:**
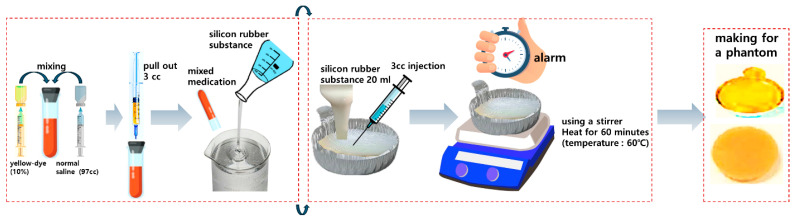
Fluorescent sodium (yellow-dye) phantom production process (mix silicone and fluorescent contrast agent and heat to create a solid phantom—enables simple measurement of fluorescence expression performance before animal testing).

**Figure 10 sensors-24-05503-f010:**
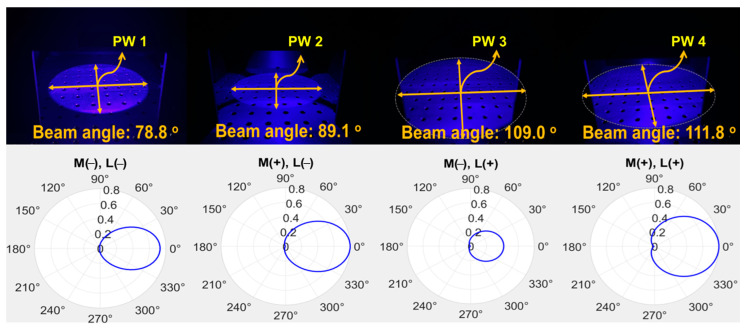
Measurement results for beam power and beam width (PW1: M(−)/L(−)—without beam mirror and concave lens; PW2: M(+)/L(−)—with beam mirror and without concave lens; PW3: M(−)/L(+)—without beam mirror and with concave lens; PW4: M(+)/L(+)—with beam mirror and concave lens), all measured for the beam width of LED illumination.

**Figure 11 sensors-24-05503-f011:**
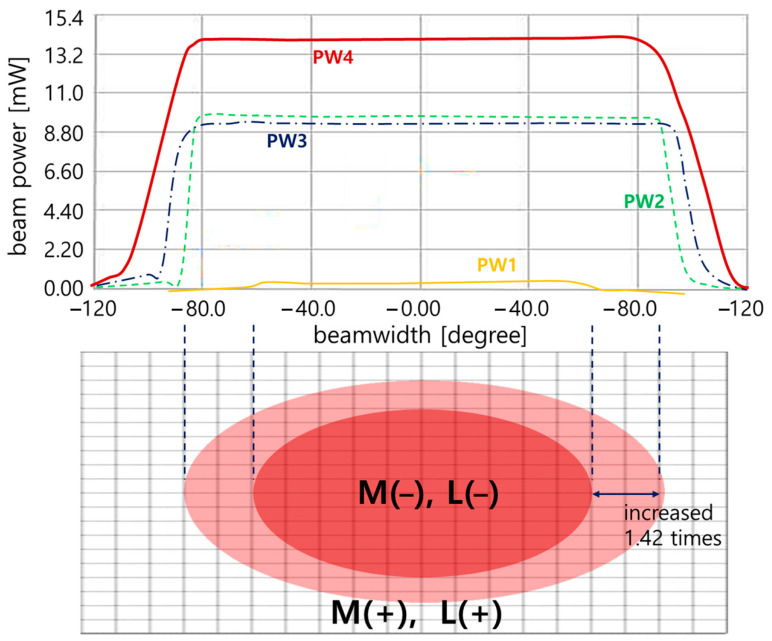
Scaling comparison analysis of beam power and beam width experiment results (PW1: M(−)/L(−)—without beam mirror and concave lens; PW2: M(+)/L(−)—with beam mirror and without concave lens; PW3: M(−)/L(+)—without beam mirror and with concave lens; PW4: M(+)/L(+)—with beam mirror and concave lens), all compared and evaluated for the beam irradiation width of the LED.

**Figure 12 sensors-24-05503-f012:**
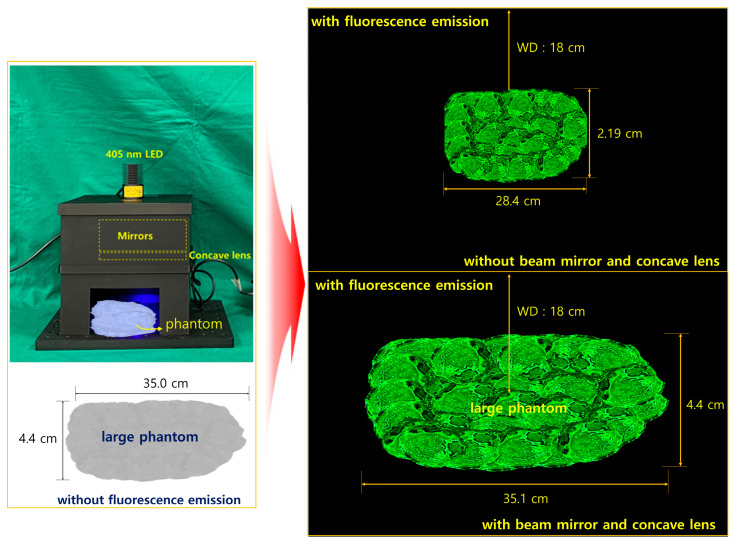
Preliminary experiment of clinical suitability of beam width and beam power through creation of a phantom similar to the lesion size and fluorescence emission, measurement of the difference in fluorescence emission area for PW2 (M(+)/L(−)—with beam mirror and without concave lens); and for PW4 (M(+)/L(+)—with beam mirror and concave lens).

**Figure 13 sensors-24-05503-f013:**
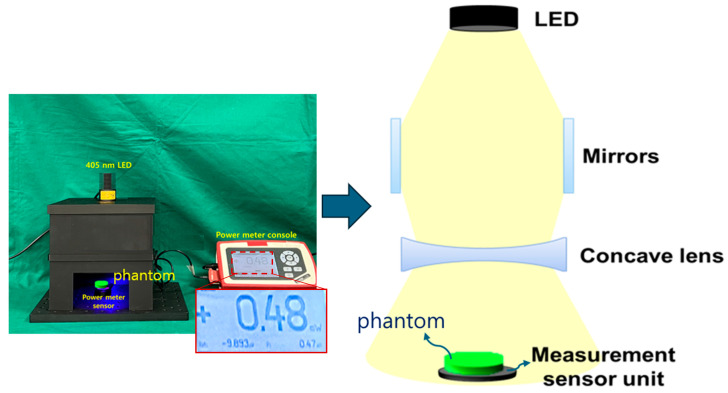
Structure of the fluorescence emission power measurements through the excitation power (mounting a fluorescence emission phantom on top of the power sensor for fluorescence emission power measurement).

**Figure 14 sensors-24-05503-f014:**
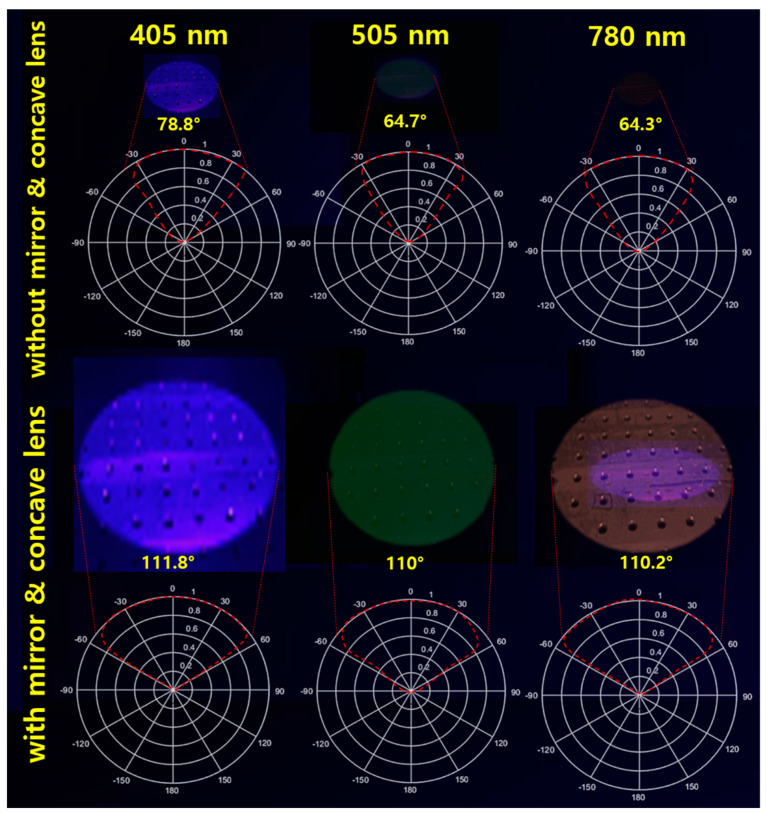
Beam width uniformity increase measurement results in various wavelength bands—comparative measurement for PW1 (M(−)/L(−)—without beam mirror and concave lens); and for PW4 (M(+)/L(+)—with beam mirror and concave lens) (Comparative measurement for LEDs dedicated to fluorescence emission (405 nm, 505 nm, 780 nm)).

**Figure 15 sensors-24-05503-f015:**
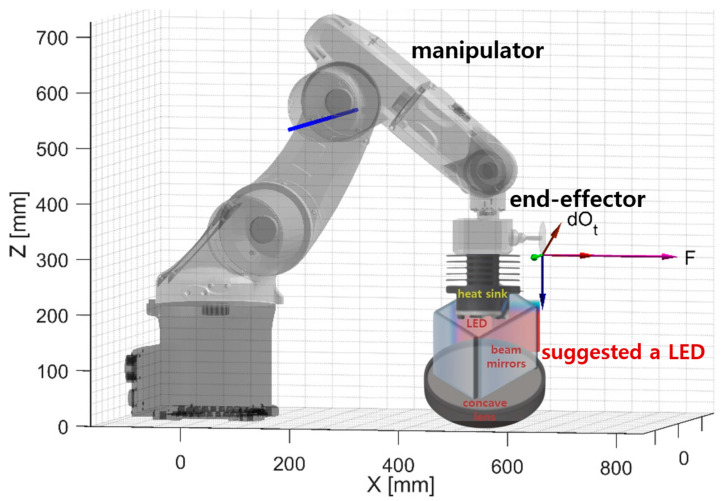
Manipulator application schematic for changing the direction angle and working distance of the LED module (both left/right, up/down direction switching and working distance can be adjusted using the end-effector).

**Figure 16 sensors-24-05503-f016:**
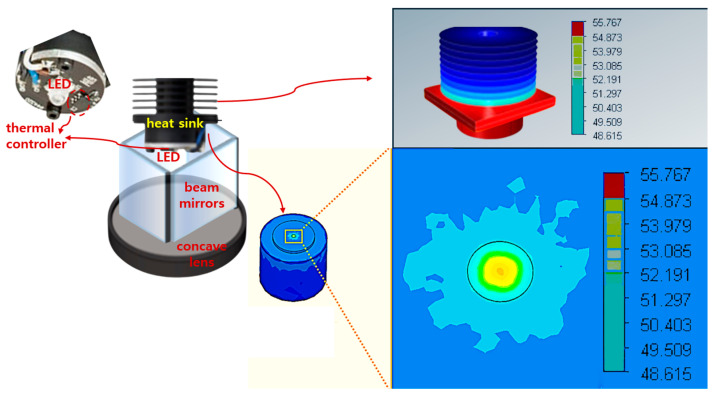
Cooling simulation results of LED modules (heat distribution test after inserting a thermal control module for LED heat control and designing a heat sink—effective cooling possible when a heat sink is installed).

**Table 1 sensors-24-05503-t001:** Summary of experiment results.

Unit	Target Power [mW]	Beam Width	Irradiation Area [cm]	Fluorescence Emission Power [mW]	Applications	
					Mirrors (M)	Concave Lens (L)
PW1	0.73	78.8°	24.8	0.12	not applicable (M−)	not applicable (L−)
PW2	10.1	89.1°	28.0	4.00	Applicable (M+)	not applicable (L−)
PW3	10.9	109°	34.2	5.30	not applicable (M−)	Applicable (L+)
PW4	13.6	111.8°	35.1	6.20	Applicable (M+)	Applicable (L+)

**Table 2 sensors-24-05503-t002:** Results of refractive amplification response in various wavelength bands.

Irradiation Wavelength [nm]	Emission Wavelength [nm]	Mirror & Concave Lens	Beam Width	Irradiation Area [cm]	Received Power [mW]	Emission Power [mW]
405	530-560	without	78.8	24.8	0.73	0.12
		with	111.8	35.1	13.6	6.20
505	550-580	without	64.7	22.7	0.69	0.11
		with	110.0	34.4	12.7	5.86
780	830-860	without	64.3	22.2	0.56	0.092
		with	110.2	34.8	11.6	5.65

**Table 3 sensors-24-05503-t003:** Comparison of the difference for suggested LED and others.

Ref. [#]	λ_ext_ [nm]	WD [cm]	Beam Width [cm^2^]	P_max_ [mW]	Target Received Power [mW]	LED Quantity [ea]
this work	405	18.0	35.1	200	13.6	1.00 (LED)
[23]	405	0.25	0.027	40.0	12.3	1.0 (laser)
[27]	550	56.75	5.72	7920	0.196	9.00 (LED)
[28]	625	30.00	3.24	300,300	26.55	130 (LED)
[29]	467	6.17	3.31	100	6.10	52 (LED)

## Data Availability

The data presented in this study are available upon request from the corresponding author. The data are not publicly available because of privacy and ethical restrictions.

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
