# Peer review of "Increasing the Beam Width and Intensity with Refraction Power Effect Using a Combination of Beam Mirrors and Concave Mirrors for Surgical-Fluorescence-Emission-Guided Cancer Monitoring Method"

_sensors, 2024, doi:10.3390/s24175503_

Round 1

Reviewer 1 Report

Comments and Suggestions for Authors

This paper centers on advancements in LED technology aimed at enhancing the beam width and irradiation power for observing color distinctions in lesions through fluorescence emission. A unique diamond-shaped beam was achieved due to specific refraction and reflection properties of the beam mirror integrated within the optical system. The study employed a concave lens to explore the refractive power effect, which effectively illuminated previously shadowed areas, thereby maximizing both beam width and intensity. This innovation facilitates comprehensive illumination of larger lesions, improving the observation of fluorescence emission and aiding in more accurate diagnostics. However, there are some issues need the authors to clarified.

    1. The authors claimed the LEDs with a mirror and concave lens have an increased beam width of more than 1.415 times, and the beam power increased by 18.6 times. While the distinction between tumor and surrounding tissues often achieved through a fluorescence signal. Therefore, the authors should determine the quantitative data on fluorescence emission enhancement.

2. It is an advantage to achieve higher beam irradiation power and wider beam width while does not require changing circuit characteristics or semiconductor processing, thus more than one LED with difference excitation wavelength should be including the experiment and evaluate the broader applicability of this method.

3. When the beam width maintained by including concave or convex lenses which also bring a disadvantage in that the light of the LED cannot be adjusted to an intended angle. The authors should have a discussion on this issue.

4. Before conducting animal experiments, more issues need to be considered. For example, the fluorescence process causes a lot of energy dissipation. In order to ensure fluorescence efficiency, higher LED light power is usually required, which will inevitably involve with thermal management and LED light stability issues. So, a long-term performance data of the LED system is necessary.

5. The caption of figures could be more detailed to increase the readability of the article.   

Comments on the Quality of English Language

 The English writing is essentially without major issues and merely needs a straightforward touch-up.

Round 1

Comments 1

The authors claimed the LEDs with a mirror and concave lens have an increased beam width of more than 1.415 times, and the beam power increased by 18.6 times. While the distinction between tumor and surrounding tissues often achieved through a fluorescence signal. Therefore, the authors should determine the quantitative data on fluorescence emission enhancement.

Answer 1

It seems that the power quantitative numerical results for fluorescence expression have been accumulated. Thank you for your important points. I added the results of my own experiments and analysis to the last text of session 3.0. Please check Table 1 and Figure 13 and the yellow part of lines 312-340.

Comments 2

It is an advantage to achieve higher beam irradiation power and wider beam width while does not require changing circuit characteristics or semiconductor processing, thus more than one LED with difference excitation wavelength should be including the experiment and evaluate the broader applicability of this method.

Answer 2

You are 100% right about the point. So we also measured using various wavelengths of LEDs that are mostly used in medical fluorescence and increased the application value. So please check the green line in the review and Figure 14 and Table 2.

Comments 3

When the beam width maintained by including concave or convex lenses which also bring a disadvantage in that the light of the LED cannot be adjusted to an intended angle. The authors should have a discussion on this issue.

Answer 3

Yes, that's right. I admit that the directional angle of the irradiated light cannot be adjusted. We focused on the beam width and beam intensity expansion effect by increasing the refractive intensity using the beam mirror and concave lens, so we did not consider the directional angle. Thank you for your good point. However, we believe that the results of our study are also meaningful. This study expanded the beam width and intensity by increasing the refractive intensity using the beam mirror and concave lens, and this result enables beam irradiation to the entire lesion and induces fluorescence emission without adjusting the beam irradiation angle. If the LED module with the beam mirror and concave lens integrated into it is connected to the end-effector using the robot's manipulator to adjust the beam irradiation angle, the beam irradiation angle and working distance can be sufficiently adjusted [10]. Therefore, there is a need for continuous research and development in the future, and these results are expected to contribute to the development of medical fluorescence observation and medical robots. These results are marked in red in Figure 15 and lines 430-439 in the discussion.

Comments 4

Before conducting animal experiments, more issues need to be considered. For example, the fluorescence process causes a lot of energy dissipation. In order to ensure fluorescence efficiency, higher LED light power is usually required, which will inevitably involve with thermal management and LED light stability issues. So, a long-term performance data of the LED system is necessary.

Answer 4

As a result of the simulation as shown in Figure 16, we found that unexpectedly, heat was generated according to your valuable advice. So we installed a heat sink and a temperature control module to efficiently manage the heat and induce it to be cooled by the heat sink, and then performed the simulation. Therefore, we found a way to efficiently manage the heat. For a detailed explanation, please refer to Figure 16 and the explanation of the blue marks in lines 444-459 at the end of the discussion. Thank you for your advice.

Comments 5

The caption of figures could be more detailed to increase the readability of the article.

Answer 5

Added detailed explanations to captions of all Figures.

Comments 6

The English writing is essentially without major issues and merely needs a straightforward touch-up..

Answer 6

Thank you. We plan to continue to review and revise/supplement the English grammar before publishing to improve the readability of the paper.

Reviewer 2 Report

Comments and Suggestions for Authors

In this paper, an LED is used as the excitation light source to excite the fluorescence of the fluorescent marker injected into the pathological area. The total reflection beam mirror and concave lens increase the beam width and excitation light intensity while a shadow area appears at the edge. However, this article only solves a small problem, and the innovation lies only in the addition of four flat mirrors, lacking innovation and wide applicability. For example, using a high-power, high divergence angle LED or multiple low-power LED arrays can also solve the problem proposed by the author.

(1) In the original text, lines 60-63 describe“The use of lasers has the disadvantages of being harmful to the human body and causing thermal destruction due to high energy consumption, resulting in many breakdowns and difficult maintenance.” Firstly, the article should limit the range of laser energy, beyond which energy can cause harm to the human body and reference. Laser-induced fluorescence detection technology has many applications in clinical medicine, such as https://doi.org/10.1016/j.snb.2022.131879. Secondly, the original text describes “resulting in many breakdowns and difficult maintenance” with no reference. This conclusion is based on the author's guess, and all data and conclusions in the Introduction should have been previously demonstrated and recommended in this article.

(2) The use of multiple LEDs to increase light intensity and beam width is a very mature technology, which can refer to the autofluorescence imaging equipment produced by Fluobeam of France.

(3) In the original text, Figure 6. of lines 186-187 which in terms of engineering, standard three views (front view, left view, and top view) should be presented. The structure cannot be intuitively reflected in the original text. The parts represented by letters in Figure 6. (a) are also not reflected in the main text or legend.

(4) In the original text, Figure 8. of lines 215-216, what is the photosensitive area of an optical power meter? What kind of filter is added in front of the probe of the optical power meter, as it does not have wavelength resolution capability? Does the intensity of the LED change during each experiment?

(5) The description of the ratio of beam width is unclear. In the original text, lines 239 and 240 describe “1.42 times”, however, in Figure 11. describe “increased 1.41 times”.

(6) In the original text, lines 259-264 describe“For the LED without a mirror, the fluorescence emission range is observed to be 24.8cm in diameter at its brightest state. Therefore, it is believed that general LEDs can only observe tumors with an area of around 35.0 cm in diameter. However, LEDs with a mirror and concave lens have an increased beam width of more than 1.415 times, so it is evaluated that the fluorescence emission can be observed in the brightest form in the range of about 35.1 cm in diameter with uniform intensity within a certain area.” The beam width increased by 1.415 times before and after using total reflection beam mirror and concave mirrors, and it was observed that the diameter of the tumor only increased by 0.1 cm ?

(7) High power LEDs also generate a large amount of heat, requiring the addition of heat dissipation fins and fans on the substrate. The overall volume and power consumption are not much different from lasers. When the output power is high, a temperature control system needs to be added to ensure the stability of the LED during use. LEDs need to age for about a week before they can achieve high stability during use. The monochromaticity of LED is not as good as Laser. Lasers of the same wavelength do not require long-term aging (about 20 minutes) when used, and there are many high-power lasers available. It can be seen from the article that the author has limited knowledge about the use of lasers in medicine.

Round 2

Comments 1

In this paper, an LED is used as the excitation light source to excite the fluorescence of the fluorescent marker injected into the pathological area. The total reflection beam mirror and concave lens increase the beam width and excitation light intensity while a shadow area appears at the edge. However, this article only solves a small problem, and the innovation lies only in the addition of four flat mirrors, lacking innovation and wide applicability. For example, using a high-power, high divergence angle LED or multiple low-power LED arrays can also solve the problem proposed by the author.

Answer 1

Thank you for pointing out the important results. Thank you for your advice. For more information, please refer to the gray lines 377-407 in the review. The method of using 4 LEDs is a groundbreaking and very good method. Of course, the unit price and power consumption will increase, and the circuit configuration method may also become complicated. However, if the LEDs are connected in parallel and the negative feedback gain increase method is used in the drive circuit, the circuit configuration will be simplified and the power consumption can be minimized. Also, if the 4 LEDs are used for rotation, it seems that the problem can be solved as you suggested. I expect that these methods will obtain positive results through future research. Thank you.

Comments 2

  • In the original text, lines 60-63 describe“The use of lasers has the disadvantages of being harmful to the human body and causing thermal destruction due to high energy consumption, resulting in many breakdowns and difficult maintenance.” Firstly, the article should limit the range of laser energy, beyond which energy can cause harm to the human body and reference. Laser-induced fluorescence detection technology has many applications in clinical medicine, such as https://doi.org/10.1016/j.snb.2022.131879. Secondly, the original text describes “resulting in many breakdowns and difficult maintenance” with no reference. This conclusion is based on the author's guess, and all data and conclusions in the Introduction should have been previously demonstrated and recommended in this article.
  • The use of multiple LEDs to increase light intensity and beam width is a very mature technology, which can refer to the autofluorescence imaging equipment produced by Fluobeam of France.
  • In the original text, Figure 6. of lines 186-187 which in terms of engineering, standard three views (front view, left view, and top view) should be presented. The structure cannot be intuitively reflected in the original text. The parts represented by letters in Figure 6. (a) are also not reflected in the main text or legend.
  • In the original text, Figure 8. of lines 215-216, what is the photosensitive area of an optical power meter? What kind of filter is added in front of the probe of the optical power meter, as it does not have wavelength resolution capability? Does the intensity of the LED change during each experiment?
  • The description of the ratio of beam width is unclear. In the original text, lines 239 and 240 describe “1.42 times”, however, in Figure 11. describe “increased 1.41 times”.
  • In the original text, lines 259-264 describe“For the LED without a mirror, the fluorescence emission range is observed to be 24.8cm in diameter at its brightest state. Therefore, it is believed that general LEDs can only observe tumors with an area of around 35.0 cm in diameter. However, LEDs with a mirror and concave lens have an increased beam width of more than 1.415 times, so it is evaluated that the fluorescence emission can be observed in the brightest form in the range of about 35.1 cm in diameter with uniform intensity within a certain area.” The beam width increased by 1.415 times before and after using total reflection beam mirror and concave mirrors, and it was observed that the diameter of the tumor only increased by 0.1 cm ?
  • High power LEDs also generate a large amount of heat, requiring the addition of heat dissipation fins and fans on the substrate. The overall volume and power consumption are not much different from lasers. When the output power is high, a temperature control system needs to be added to ensure the stability of the LED during use. LEDs need to age for about a week before they can achieve high stability during use. The monochromaticity of LED is not as good as Laser. Lasers of the same wavelength do not require long-term aging (about 20 minutes) when used, and there are many high-power lasers available. It can be seen from the article that the author has limited knowledge about the use of lasers in medicine.

Answer 2

  • It was claimed that only lasers are harmful to the human body and generate heat, causing many malfunctions. Come to think of it, even high-power LEDs generate a lot of heat. Therefore, I analyzed the scope of application of the 405 nm wavelength band in medicine and added references. LEDs and lasers are not the only ones to pay attention to. It was analyzed that 405 nm is harmful to the human body and is used in the medical field for sensors, diagnosis, and sterilization. Please check the blue lines 444-459 at the end of the review.
  • Fluobeam is a device that uses indocyanine green fluorescent contrast agent to observe blood circulation and lymph nodes in blood vessels, and it is a very excellent device. However, it is regrettable that it uses multiple LEDs, is expensive, and has a complicated circuit configuration. A detailed explanation is indicated in red sentences (lines 394-407) in the review gray. Please refer to it.
  • I modified it. Please refer to the pink lines 186-188 and Figure 6 in Session 3.
  • The LED intensity changes. The power reaching the power sensor is 13.6 mW. This 13.6 mW power is when the optical bandpass filter is installed. If the optical filter is not applied, the power reaching the power sensor is approximately 13.72 mW. That is, there is a difference of 0.12 mW. For details, please refer to the gray (lines of 221-234) in session 3 and Figure 8 (b). Thank you.
  • I made a correction. Thank you very much for your detailed comments. 1.42 times is correct and 1.41 times is a typo. Thank you again.
  • I think I made a mistake in the expression during the manuscript writing process. I sincerely appreciate your comments. So I revised the explanation again. Please check the blue sentences at the bottom of Figure 12 (lines of 306-311). Thank you.
  • I think I may have been overly sensitive in the negative treatment of lasers in the medical device approval process. Thank you for your advice. After reanalyzing your advice, I realized that high-power LEDs generate heat. The advice on aging for a week for LED stability (laser for about 20 minutes) and the advice on monochromatic LEDs being harmful to the human body seem to be quite helpful. Therefore, I removed the sentences about lasers being harmful to the human body and about heat. In addition, I simulated the heat management of high-power LEDs as shown in Figure 16. For more information, please refer to the lines 444-459 in the light blue section.

Round 2

Reviewer 1 Report

Comments and Suggestions for Authors

None

Comments on the Quality of English Language

None

Author Response

Answer 1

Thank you for your evaluation of my paper. I will improve the quality of my paper through proofing version before publishing.

Reviewer 2 Report

Comments and Suggestions for Authors

It is recommended that the author carefully review and revise the meanings represented by all abbreviations and symbols in the article before publishing.

Comments on the Quality of English Language

The original symbols represent unclear meanings, such as "dj-n" and "djn" in line 114.

Author Response

Comments 1

The original symbols represent unclear meanings, such as "dj-n" and "djn" in line 114..

Answer 1

Thanks for the pointers on the symbol. I've corrected it correctly. Please check the yellow text on line 114.
